ecology

population viability, wind energy, raptor, Black Harrier, mortality, citizen science

**Author for correspondence:**
Francisco Cervantes
e-mail: f.cervantesperalta@gmail.com

# Population viability assessment of an endangered raptor using detection/ non-detection data reveals susceptibility to anthropogenic impacts

Francisco Cervantes[1,2], Marlei Martins[3] and Robert E. Simmons[2,3]

[1]Centre for Statistics in Ecology, Environment and Conservation, and [2]FitzPatrick Institute of African Ornithology, University of Cape Town, Rondebosch 7701, Cape Town, South Africa
[3]Birds and Bats Unlimited Cape Town, South Africa

FC, 0000-0001-7189-4070; RES, 0000-0003-4964-8126

As the demand for carbon-neutral energy sources increases, so does the need to understand the impacts that these technologies have on the environment. Here, we assess the potential consequences of additional mortality on an Endangered raptor recently exposed to wind farms for the first time, the Black Harrier *Circus maurus*, one of the world's rarest harriers. We conduct a population viability assessment using a Bayesian model integrating life-history information and annual reporting rates from detection/non-detection surveys from the South African Bird Atlas Project. Our model estimates a global population of approximately 1300 birds currently declining at 2.3% per year, and one that could collapse in under 100 years, if an average of three to five adult birds are killed annually. This level of mortality may soon exist, given the current rate of fatalities and the number of wind farms planned within the species' distribution. In addition, we find that the population is sensitive to changes in climate. Our results highlight the critical need for appropriate placement, and adaptive management of wind farms and other infrastructure causing harrier mortality. We also show how detection/non-detection data may be used to infer population dynamics and viability, when population counts are unavailable.

# 1. Introduction

Impact on bird populations is one of the main environmental concerns associated with the growing expansion of wind energy [1]. While wind energy production is responsible for fewer overall fatalities than other human activities, its impacts on long-lived species with low reproduction rates, such as raptors and other large soaring birds, has motivated scrutiny all over the world [2,3]; more so when the affected species are endangered or range-restricted. Considerable efforts have been devoted to quantifying and monitoring the number of bird casualties produced by wind energy facilities, as well as to predicting collision risk (e.g. [3–7]). However, it is critical to go one step further and contextualize these numbers to understand the repercussions of this additional mortality on bird populations, and investigate whether these impacts are sustainable over time [8].

With the implementation of the Renewable Energy Independent Power Producers Procurement Programme, just under 3.5 gigawatt of wind energy have been installed in South Africa between 2010 and 2020 [9]. Renewable energy development is planned to continue to grow and replace the ageing energy-production infrastructure in the country. Meanwhile, South Africa boasts one of the world's biodiversity hotspots [10] and significant efforts are in place to conserve these natural values, while boosting economic development [11]. Among these biodiversity riches are 79 species of raptors of which 25% are classified as threatened in South Africa [12]. We focus our work on the Black Harrier *Circus maurus*, one of the world's rarest harriers [13]. The species, endemic to Southern Africa, with *ca* 99% of its population in South Africa [14], exhibits very low genetic diversity [15] and has undergone a long-term reduction in numbers due to habitat loss over the last century [12,16]. It is therefore important to predict the consequences of impacts associated with human activity on this already fragile species.

Elsewhere in the world, the effect of wind energy facilities on other harriers (*Circus* spp.) appears to have been relatively minor (see [17] for North American Northern Harriers *Circus hudsonius* and [18–20] for the three European species). The reason why relatively few harrier fatalities have been reported could be that these species show particular caution around turbines [17,21] or that breeding birds tend to move away from operational wind farms [20,22,23]. However, in the secondary literature there are over a hundred mortality records of European harriers, putting them in the top 25% of raptors experiencing wind farm deaths [24–26], and highlighting the need for further research.

With its small, range-restricted population, the Black Harrier could be impacted by both mortality and displacement. Mortality produced by wind farms is particularly relevant for the Black Harrier because, traditionally, the main cause of decline of its population has been habitat destruction, and not direct mortality associated with human activities [27]. This additional mortality could add a new dimension to the management of this species. To investigate the magnitude of these potential impacts and to inform appropriate conservation measures, we conduct a population viability assessment. We use a Bayesian model that uses detection/non-detection data and life-history information for the species to infer the rate of change of the population [28,29]. Using detection/non-detection data from the Southern African Bird Atlas Project (SABAP2, [30]), we overcome the need of having detailed demographic data, which is one of the main obstacles when conducting a population viability assessments [31,32]. We estimate population size and life-history parameters simultaneously using information published on the breeding ecology and satellite-tracked movements of the Black Harrier to define sensible priors for the model [14,16,33–35]. Integrated population modelling provides important benefits in terms of the number of parameters that can be estimated, as well as in the precision of the estimates [36].

Thus, our aim here is twofold: (i) to investigate the use of detection/non-detection data to infer dynamics of wildlife populations affected by human infrastructure, and (ii) to study the viability of the Endangered Black Harrier, providing urgent quantitative guidance on how many additional losses this geographically restricted population can take, based on current knowledge.

# 2. Methods

## 2.1. Population dynamics model

To understand the effect of additional mortality on Black Harrier populations, we first defined a model for the current population dynamics of the species, and then we simulated population trajectories under different levels of additional mortality.

To describe the current population size and rate of change for the species, we fitted a Bayesian dynamic extension of a Royle–Nichols model to detection/non-detection data [28,29,37]. Data on detection/non-detection of Black Harriers throughout South Africa was extracted from the Southern African Bird Atlas Project (SABAP2, [30]). This is a citizen-science project, in which participants produce lists of observed bird species during visits to a grid of pentads ($5 \times 5$ min cells). Pentads are sampled each year on several occasions, recording detection or non-detection of the species of interest. The number of pentads, the individual pentads visited and the number of visits to each pentad may change from year to year. Species' lists are used to delineate the distribution of the different bird species. Note that we purposefully avoid referring to presence/absence, because non-detection does not necessarily imply absence, due to observers' imperfect detection skills. For each year, we used those pentads visited at least five times, throughout the distribution range of the Black Harrier.

The basic model by Royle & Nichols [29] builds on the fact that detection of a species at a site (pentads in our case) depends on the abundance of that species. More specifically, if $r$ is the probability of observing any one individual, then the probability $p_i$ of observing at least one individual in site $i$ is

$$p_i = 1 - (1 - r)^{N_i},$$ (2.1)

where $N_i$ is the number of individuals at site $i$.

We gave the number of individuals across $M$ different sites in any given year a negative binomial ($\mu$, $\psi$) distribution. The extra parameter $\psi$ of the negative binomial (in relation to the Poisson used by Royle & Nichols [29]) accounts for over-dispersion in the data, relaxing the equality condition between mean and variance. Note that in this specification of the negative binomial, the variance is $\mu + \mu^2/\psi$ [38].

The reporting rate of the species of interest at site $i$ (number of detections at site $i$/total number of visits to site $i$) in year $t$ is modelled as a binomial process that is conditional on the (unobserved) number of individuals on this site. The likelihood of the number of detections at site $i$ and year $t$ is

$$\mathcal{L}(d_{i,t}) = \sum_{k=0}^{K} \text{binomial}(d_{i,t}|J_{i,t}, p_k) \times \text{neg. binomial}(k|\mu_t, \psi).$$ (2.2)

The subscript $t$ indexes the year when observations were made, $d_i$ is the number of detections at site $i$, $k$ is the number of individuals at site $i$, $J_i$ is the total number of visits to site $i$, $p_k$ is the probability of detecting at least one individual, provided that there are $k$ individuals present, and $\mu_t$ is the mean number of individuals across sites in year $t$.

Theoretically, it would be necessary to consider any number of birds per site and the summation should go to infinity, but in practice, we may set an upper limit $K$ for the average number of individuals across sites. In this case, we set an upper limit of 10 birds per pentad, based on past experience from Black Harrier surveys. We also tested a maximum number of five birds on average and the results were nearly identical.

Once we have a model for the size of the population of Black Harriers in any given year, we define the population rate of change, denoted by $\lambda_t$. Considering that the mean abundance across pentads in year $t$ is $\mu_t$, we define $\mu_{t+1} = \lambda_t \mu_t$. We now proceed to incorporate all we know about the life history of the Black Harrier to model $\lambda_t$.

## 2.2. Linking population trend with life-history parameters

We structure the life history of the Black Harrier, defining three age classes: fledglings—individuals that have just left the nest, sub-adults—for individuals that are one year old, and adults—for individuals two years old and older. Adult individuals are the only birds capable of breeding. In addition to an age structure, we defined survival rate as the probability of survival in a given year, and fecundity as the number of offspring per adult bird per year (note that fecundity usually refers to offspring per female, but we have re-defined this to simplify notation later).

We represent the population of harriers at year $t$ as a column vector $\mathbf{N}_t = [n_{0,t}, n_{1,t}, n_{2,t}]^{\mathrm{T}}$, where $n_{0,t}$ represents the number of fledglings, $n_{1,t}$ the number of sub-adults and $n_{2,t}$ the number of adults in the population. The change in number of harriers from year $t$ to year $t+1$ is characterized by the transition matrix $\mathbf{A}_t$ such that

$$\mathbf{N}_{t+1} = \mathbf{A}_t \mathbf{N}_t$$ (2.3)

where

$$\mathbf{A}_t = \begin{bmatrix} 0 & \rho_t \phi_2 & \rho_t \phi_2 \\ \phi_1 & 0 & 0 \\ 0 & \phi_2 & \phi_2 \end{bmatrix}, \tag{2.4}$$

where $\rho_t$ and $\phi$ represent fecundity and survival rate, respectively. As a reminder, we defined fecundity as the mean number of offspring per individual in a year, and survival as the proportion of birds that survive a given year on average. The subscript 1 indicates the parameter is associated with fledglings and the subscript 2 that is associated with either sub-adults or adults. Note that once the population matrix is laid out, it becomes evident that the sub-adult and adult columns are identical and therefore redundant. A population matrix with only adults would convey the same information. We decided to leave the three-age-class structure to prove that removing adults or sub-adults from the population has the same effect.

To inform fecundity values, we followed García-Heras *et al.* [33] who observed that 31% of 223 monitored Black Harrier nests failed to produce any offspring and that the average number of chicks produced by a nest was 1.65 (s.d. = 1.30, $n = 222$). The maximum number of offspring observed was four. García-Heras *et al.* [33,35] showed that Black Harrier fecundity is dependent on rainfall and therefore, we incorporated rainfall as a covariate for fecundity. More precisely, we identified all weather stations in a 400 km radius of Laingsburg (Latitude: $-33.2$, Longitude: 20.9) and calculated the daily average rainfall across stations. We used the data provided by the Global Historical Climatology Network (www.ncdc.noaa.gov), retrieved using the R package `rnoaa` [39]. We used a daily average to deal with the different number of days each station had data for. We modelled fecundity as,

$$\rho_t = 1.5 \, \text{logit} \, (\beta_0 + \beta_1 \text{rain}_t + \gamma_t)$$
$$\gamma_t \sim \text{normal}(0, \sigma_\gamma). \tag{2.5}$$

With the logit transformation above, we ensure that mean fecundity per harrier stays within zero and 1.5 birds (i.e. three birds per pair). The term $\gamma_t$ represents a random effect affecting fecundity each year. Adding a random effect to the fecundity parameter is a way of introducing certain flexibility in the number of harriers in the population in each year. Although it does imply that random changes in the population are due to changes in fecundity, it circumvents the need for adding random terms to the number of individuals per age class directly, which, with our limited data, caused model instability and non-identifiability of the parameters. We used accumulated rainfall in the south west of South Africa, because this is where the core of the Black Harrier breeding population occurs [14,16,33].

To inform our estimates on adult survival rates, we fitted an exponential survival curve to observations from [34], who confirmed three fatalities out of 13 tracked breeding birds (tracked for an average of 365 days). We added a fourth bird that was confirmed dead by RES but was not accounted for in [34], although the addition of this fourth bird did not affect model results substantially.

We have no information with regards to juvenile survival of Black Harriers, so we estimated adult survival ($\phi_2$) from the tracking data and assumed lower survival rates for birds in their first year than for older birds, as found for other species of harriers from Europe ([40–42]). We therefore constrained the parameter $\phi_1$ to be less than $\phi_2$ (see model in electronic supplementary material, S2). We considered that mean survival rates are constant over time.

We fitted the model within a Bayesian framework estimating the exponential survival curve, annual fecundity and probability of detection jointly [36]. For a more detailed explanation of and code for fitting the model, refer to the electronic supplementary material.

We used R to run the analysis [43] with the added functionality of the `tidyverse` packages [44]. We wrote the model in `Stan` and used the package `rstan` to work from R [45]. We ran four Hamiltonian Monte Carlo chains of 2000 iterations, discarding the first 1000 as adaptation.

## 2.3. Mortality simulations

To investigate the behaviour of Black Harrier populations under different scenarios of additional mortality, we ran 1000 Monte Carlo simulations of 100-year-long population trajectories, sampling life-history parameters from the posterior distribution of the fitted population dynamics model. Since fecundity depends on rainfall, for the simulations we sampled rainfall values from the data with replacement, at each step of the simulation.

We repeated the simulation process using different levels of additional mortality: 0, 1, 3 and 5 adult individuals per year removed from the population. The simulation process was as follows:

1. Define the desired level of mortality for each year.
2. For each trajectory, sample an initial population size and life-history parameters from the estimated posterior distribution.
3. Within each trajectory, sample 100 rainfall values from those in the data and calculate fecundity for each simulated year.
4. Within each trajectory, and using a step size of 1 year, update the number of harriers in the different age categories according to the parameters sampled in step 2 and fecundity calculated in step 3.
5. Subtract the number of harriers defined in step 1 from the population resulting in step 4.
6. Repeat steps 4 and 5 until 100 years have been simulated.

We repeated the above process removing fledglings (first year birds) and sub-adults instead of adults. In addition, to understand what stages in the life history of the harriers have the greatest impact on the dynamics of the population, we analysed the elasticities associated with the simulated transition matrices [46,47]. Elasticities may be interpreted as proportional partial derivatives with respect to changes in different life history stages of the harriers. Although this is only true for populations that are in a steady state, the distribution of elasticities at each step should provide a reasonable, easy-to-compute approximation of the elasticities of the harrier population over the study period. We computed the probability of extinction each year as the proportion of the trajectories that reached zero in relation to the total number of simulated trajectories.

# 3. Results

Between 2008 and 2019, Black Harriers were detected in 1070 pentads (figure 1). Of these pentads, an average of 588 pentads were visited each year (s.d. = 145). The mean number of visits to each pentad each year was 6.35 (s.d. = 11.8), although the distribution was right-skewed (median = 2). After filtering the data to those pentads visited at least five times in a year, we worked with 2550 visits to an average of 212.5 pentads per year (s.d. = 35.39).

$\hat{R}$ values close to 1 indicated model convergence and parameters were estimated with a large number of effective samples (see [48], and table 1). We assessed model fit by running posterior predictive simulations of the probability of observing at least one Black Harrier in a pentad and comparing it to the observed reporting rates (figure 2). Based on this analysis, we concluded that the model captured the general structure in the data. A pronounced increase in reporting rates in the year 2014 fell out of a 95% posterior prediction credible interval, resulting in a coverage of 91.7% (lower than expected). The coverage of the 90% prediction credible interval was 91.7%, matching our expectations.

The posterior mean adult survival rate was 0.704 ± 0.044 (s.d.), and the posterior mean fledgling survival 0.502 ± 0.070 (table 1). The posterior mean fecundity was 0.783 ± 0.164 fledglings per adult harrier if rainfall is set to the average value observed during the study period. Fecundity was strongly affected by rainfall (see $\beta$ parameters in table 1, and figure 3). As an indication, an increase of one standard deviation in mean daily rainfall (ca 0.37 mm) translated to an increase of approximately 38% in fecundity. These life-history parameters (survival rates and fecundity) produced a mean population rate of change of 0.977 ± 0.132 (i.e. approx. 2.3% annual decline). The model also estimated an initial population of 1.277 ± 0.140 Black Harriers per pentad on average. Considering that the species was detected in 1070 pentads between 2008 and 2019, and simulating from the posterior distribution of mean initial abundance ($\mu_0$), we obtain an estimate of 1367 ± 158 Black Harriers in the population of 2007. Less sensitive to the initial population estimate is the posterior mean of the population during the study period, which was 1306 ± 80.

The elasticity analysis showed that changes in adult survival have a greater impact on population changes (elasticity = 0.564 ± 0.056) than fecundity (elasticity = 0.220 ± 0.031) or fledgling survival (elasticity = 0.220 ± 0.031).

Our simulations revealed that, given our model, the probability of extinction of the species in under 100 years increases substantially with each individual removed from the population (figures 4 and 5). Under our model and considering similar rainfall conditions to those observed during the study period, we expect the Black Harrier population to endure another 100 years. With one additional adult being removed from the population per year, the species could go extinct in under 100 years with probability 0.23. With three adult individuals removed from the population per year, the

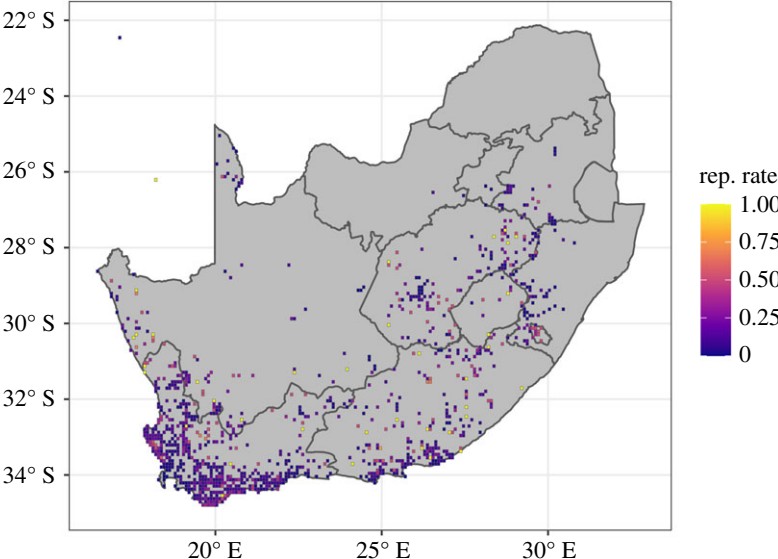

**Figure 1.** Southern African Bird Atlas Project pentads covered in this study, and the overall Black Harrier reporting rate (number of detections/number of visits) for the period 2008–2019.

**Table 1.** Posterior parameter distribution estimated for the population dynamics model. The table shows the posterior mean, standard deviation, 95% credible region (2.5–97.5%), median (50%) as well as the number of MCMC samples from the posterior (n_eff) and the convergence diagnostic $\hat{R}$ (values close to 1 indicate convergence).

| parameter | mean | s.d. | 2.5% | 50% | 97.5% | n_eff | Rhat |
|---|---|---|---|---|---|---|---|
| fledgling survival ($\phi_1$) | 0.502 | 0.070 | 0.362 | 0.503 | 0.630 | 3475.321 | 1.000 |
| adult survival ($\phi_2$) | 0.704 | 0.044 | 0.624 | 0.702 | 0.796 | 1596.518 | 1.003 |
| Pr. detection ($r$) | 0.057 | 0.002 | 0.053 | 0.057 | 0.061 | 3884.898 | 1.000 |
| initial mean abundance ($\mu_0$) | 1.277 | 0.140 | 1.020 | 1.270 | 1.578 | 2527.397 | 1.000 |
| logit mean fecundity ($\beta_0$) | 0.092 | 0.459 | −0.813 | 0.088 | 1.011 | 2251.070 | 1.004 |
| logit rain effect ($\beta_1$) | 0.862 | 0.285 | 0.344 | 0.844 | 1.454 | 3312.337 | 0.999 |
| logit random fecundity ($\sigma_\gamma$) | 0.338 | 0.320 | 0.013 | 0.266 | 1.068 | 2424.401 | 1.000 |
| over-dispersion ($\psi$) | 1.365 | 0.115 | 1.153 | 1.362 | 1.603 | 4095.914 | 1.000 |

probability of extinction increases to 0.61. Finally, with five adults removed per year the estimated probability of extinction in 100 years is 0.75. An identical effect was observed if sub-adults were removed from the population, but lower extinction probabilities were estimated if fledglings were removed (see electronic supplementary material, S3).

## 4. Discussion

Our population viability assessment indicates that Black Harriers are remarkably susceptible to additional mortality, more so if it affects adult birds. Although we focused on wind energy, because it is a new threat to the species, our results could be viewed in the context of any other additional mortality source. Our baseline population status derived from a decade of bird atlas data revealed an average decline of 2.3% per annum, a trend that alone can result in extinction in just over 200 years. The analysis of Black Harrier population under additional mortality reveals that time to extinction is significantly reduced with every adult individual removed from the population. As such, an additional mortality of five adult birds killed per year could result in the global population collapsing in under 75 years with 50% probability. A similar effect is expected if sub-adult individuals are removed from the population instead of adults. The wide credible intervals in population trajectories were expected, and are indicative of the uncertainty in the true population trend inferred from the

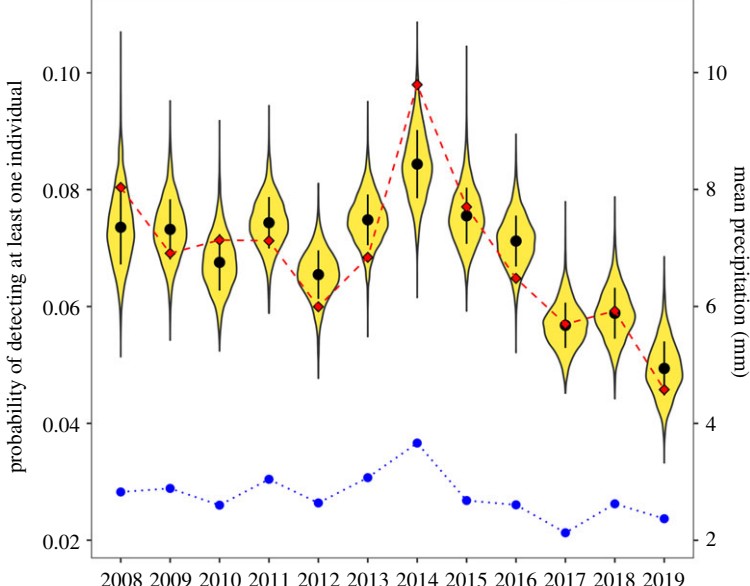

**Figure 2.** Posterior predictive distribution of the mean probability of observing at least one Black Harrier across pentads between 2008 and 2019. The violin plots show the distribution of values predicted from the posterior distribution of the population dynamics model and the black points with error bars represent the mean ± 1 s.d. of these predicted values. The red diamonds connected by the dashed red line represent the actual observed reporting rates. The blue points connected by the dotted line represent mean daily rainfall across stations.

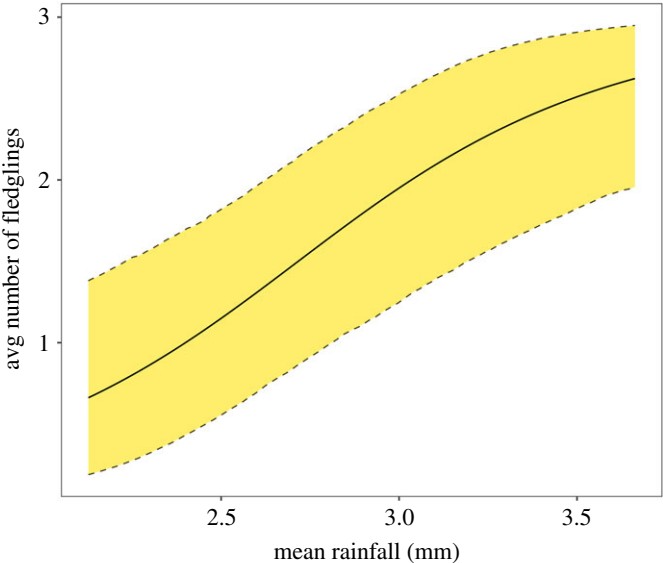

**Figure 3.** Posterior mean and 95% credible interval of the number of fledglings expected per pair of Black Harriers, conditional on daily average rainfall across weather stations in the year.

detection/non-detection data. This is particularly true at far time horizons, such as 100 years, which was chosen long enough to be able to estimate extinction probabilities. The most important point is that a large portion of the posterior predictions show a decreasing population, more so as more individuals are removed.

We considered a constant extraction of birds every year, which means that survival rate varies with fluctuations in the population (the ratio extraction-to-population changes). While a density-dependent mortality could be considered more realistic, there is no evidence for density-dependent population parameters in harrier species [13]. Also, it is not clear how harriers may react to the presence of new

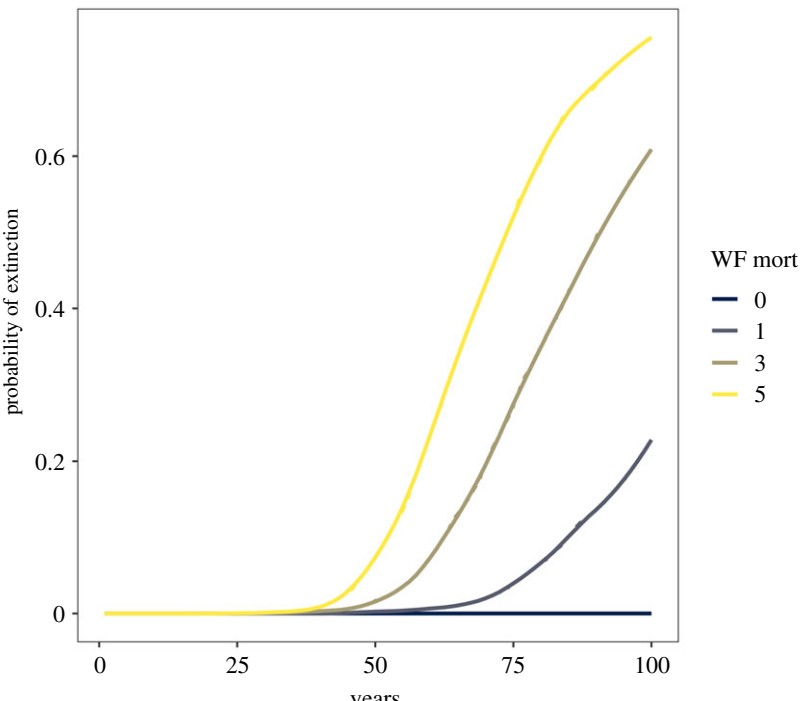

**Figure 4.** Distribution of Black Harrier population trajectories simulated from 1000 sets of life-history parameters sampled from the posterior distribution of the population dynamics model and under increasing numbers of adult Black Harrier fatalities per year (0, 1, 3, 5). The solid line represents the mean, and the brown and yellow areas represent the 50% and 90% credible regions, respectively.

**Figure 5.** Probability of extinction over a 100-year period, calculated as the proportion of simulated population trajectories (out of 1000) that reach zero each year, under increasing number of adult Black Harrier fatalities (0, 1, 3, 5). The solid lines are smoothed versions of the underlying dashed trajectories, the agreement is such that it is difficult to differentiate between the two.

infrastructure, and avoidance behaviour may blur any density dependent effects. Therefore, we considered a constant extraction scenario to be more illustrative for management purposes. We do recognize that this further implies no behavioural adaptation of the species to the new mortality source, which is something that remains unclear.

We estimate a population of *ca* 1300 Black Harriers, although this number is an approximation subject to model assumptions. Black Harriers are known to move between breeding and non-breeding areas [14,34], and therefore, harriers could be detected twice: once during breeding and once during the non-breeding period, resulting in an over-estimation of population size. In addition, a model for abundance based on detection/non-detection data can give biased estimates for rare species because of their low detection probabilities [29]. Nevertheless, our estimate is in line with previous ones [49], and the population rate of change, which is our main focus, should be independent of population size in the absence of density dependence, and thus, less sensitive to the assumption of a non-migratory population.

Overall, the parameters estimated by our model agree with previous knowledge about the Black Harrier, which reinforces our confidence both in our predictions and in our understanding of the ecology of the species. The model estimates a survival rate for adult Black Harriers around 0.7 and for fledglings of about 0.5. These numbers are similar to those estimated for other harrier species [40,41]. Mean fecundity per pair is estimated to be 1.51 fledglings (0.75 per breeding individual), which is similar although somewhat lower than the 1.9 previously observed by Curtis *et al.* [16] and slightly less than the 1.65 reported by García-Heras *et al.* [35]. Slightly lower estimated fecundity could be produced by the assumption in our model that all adult breeding pairs attempt to breed, which might not be realistic, because some pairs take sabbatical years [34]. However, we feel that this decrease may arise because of the close link that is emerging between harrier fecundity and rainfall uncovered by García-Heras *et al.* [35] and confirmed in this study. If this was the case, changes in rainfall regimes could impact our predictions, because our population projections considered a scenario where future rainfall remains at levels observed during the study period (2008–2019). Climate analyses suggest that the rain season might be shortening in the Western Cape of South Africa and that the total rain might be decreasing [50,51], which could result in a steeper population reduction than anticipated by our model.

We know that at least eight Black Harriers were killed by wind farms in South Africa between 2016 and 2020 [7]. Considering that Black Harriers are rare and have core breeding areas in the south Western Cape, where wind farms are proliferating, the mortality figures presented in this study are grounded in present rates of harrier mortality and projections of wind farm development in South Africa [7]. While there is evidence showing that some harrier species tend to move away from wind farms [20,21,23,52], breeding Black Harriers do not appear to exhibit this behaviour (R.E. Simmons, M. Martins and O. Curtis 2021, unpublished data), which may partly explain the observed mortality rates. A study of Black Harriers breeding within an Eastern Cape wind farm revealed that harriers are more susceptible to mortality during breeding, as the proportion of time spent flying within the blade swept area rose from 0 to 45%, and so coincided with more harrier deaths [53]. In addition, the species is known to perform very wide-scale movements and hold large home ranges, which may expose them to numerous wind farm environments (breeding birds move on average 16.4 km from their nests, with estimates of home-range size of $92.7 \pm 66.6$ km$^2$ and $147.8 \pm 205.4$ km$^2$ for breeders and non-breeders, respectively: [34]).

It is unclear how Black Harriers will react to increasing pressure by wind farms and other infrastructure. However, given that the species already suffers from low genetic diversity and habitat loss throughout its restricted distribution [14–16], it is unlikely that it has the mechanisms or latitude to absorb additional mortality. Furthermore, due to the diminishing extent of favourable Black Harrier habitat [16], we anticipate that vacant territories produced by additional mortality could be readily occupied by floating individuals, which could in turn create a population sink.

While in this study we have not analysed how population parameters vary spatially, it is apparent from the existing literature that coastal territories are more productive and more frequently used than inland sites [16,33,35]. Therefore, it could be argued that mortality at coastal regions would have greater impacts on Black Harrier populations than inland. However, our model reveals an exquisite sensitivity of Black Harrier populations to adult survival rates. This is consistent with previous studies that showed population of long-lived birds being largely influenced by adult survival [54–56]. Thus, given the sensitivity of the population to adult survival, and less so to reproductive indices, mortality risk should be considered similarly detrimental to the global harrier population, regardless of the geographical location. Although temporally explicit models (e.g. using Bernoulli likelihood to model detections) could be preferred for some applications, preliminary tests indicated little gains to justify the extra complexity, in our case.

There were 22 operational wind farms in South Africa in 2020 and many more are proposed to reach South Africa's renewable energy targets. Therefore, it is vital for the species that wind farms are constructed away from its breeding habitats and that Black Harrier populations are monitored closely to verify the accuracy of our model predictions and add relevant details about the ecology of the species where needed. We hope that the Black Harrier monitoring guidelines published by Simmons *et al.* [53] promote the collection of relevant data on the Black Harrier to make wind energy development compatible with the conservation of the species.

## 5. Conclusion

Our study modelled the effects of additional fatalities of Endangered Black Harriers on the global population of this rare endemic. We focus on wind energy development, because it is a new threat to the species. Some fatalities have already been recorded at wind facilities, and our modelling showed some alarming trends that highlight why it is critical to go beyond simply recording the number of fatalities. At mortalities rates of just three to five adult birds per year, the Black Harrier population could crash in under 100 years. This striking result highlights the need to avoid potentially harmful infrastructure within the breeding range of the Black Harrier that could impact this fragile population. It additionally highlights the need to model other small susceptible populations living on the edge under threat from multiple anthropogenic pressures. We have shown how detection/non-detection data can provide valuable numeric assessments about the effects additional mortality have on wildlife populations, extending the range of tools available to environmental impact assessment practitioners and management agencies.

Data accessibility. All scripts used in this study are openly accessible through https://github.com/StochasticBiology/boolean-efflux.git. The data are provided in electronic supplementary material [57]. The paper does not use new data, but it does use previously published data. Instructions on how to access the data, as well as all the code necessary to run the analysis, are detailed in the electronic supplementary materials, S1 and S2.

Authors' contributions. F.C.P.: conceptualization, formal analysis, methodology, project administration, writing—original draft, writing—review and editing; M.M.: conceptualization, writing—original draft, writing—review and editing; R.E.S.: conceptualization, data curation, methodology, writing—original draft, writing—review and editing.

All authors gave final approval for publication and agreed to be held accountable for the work performed.

Competing interests. We declare we have no competing interests.

Funding. National Research Foundation incentive grant no. 90582 to R.E.S.

Acknowledgements. Our analysis builds upon Black Harrier research by Drs Sophie Garcia-Heras, Bea Arroyo, Francois Mougeot, Odette Curtis and Andrew Jenkins, and we thank them for their advice. We also thank Prof. Res Altwegg for his valuable inputs on modelling population dynamics. Drs Phil Whitfield, Ralph Buij and Tonio Schaub kindly answered our questions about European harriers and wind farms. We also thank two anonymous reviewers for providing important feedback that enhanced the paper. We acknowledge Golden Fleece Merino, Birdlife South Africa bird clubs, NRF (incentive grant no. 90582) and the University of Cape Town for their funding to data collection by RES over the years. Thanks also to Globeleq (P. Oosthuizen) for funds to undertake studies at the Jeffreys Bay wind facility, and to citizen scientists who contributed to SABAP2.

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
