## [Peer Review File · Royal Society Open Science]

Review History

RSOS-210599.R0 (Original submission)

Review form: Reviewer 1 (Alan Lee)

Is the manuscript scientifically sound in its present form?

No

Are the interpretations and conclusions justified by the results?

Yes

Is the language acceptable?

Yes

Do you have any ethical concerns with this paper?

No

Have you any concerns about statistical analyses in this paper?

Yes

Recommendation?

Major revision is needed (please make suggestions in comments)

Comments to the Author(s)

Black Harriers are Endangered raptors that are endemic to southern Africa. This paper examines the population viability given inevitable wind infrastructure developments across the range. Use is made of the SABAP2 citizen science project. The use of this data to conduct a PVA is especially novel. The paper is topical, current, well written, and I hope it has the intended impact of encouraging conservation of this beautiful bird of prey. While framed in the light of potential wind energy impacts, realistically one could replace wind energy with any negative impacts.

I base my review on pages 14 - 25 of the pdf provided, since this seems to include in text references etc, the previous pages might be an old version, and the purpose of it being rendered in the pdf was not clear to me. I also do not like numeric citation style for the purposes of review (completely unreasonable to follow each link), and so I make no comment on the pertinence of all selected in-situ references, which I could have done with Author Date style: that is critique is on the journal. The few references I checked seemed fine.

The extent of the supplementary information is extraordinary and exemplary in terms of tidiness and succinctness: it is beautiful code. However, it fails reproducibility. I managed to reproduce code given the data provided until the stan model, when things failed and I was unable to proceed despite much time spent at attempting debugging on my side. This was the persistent error message:

SYNTAX ERROR, MESSAGE(S) FROM PARSER:

Duplicate declaration of variable, name=mu; attempt to redeclare as vector in transformed parameter; previously declared as real[] in transformed parameter

error in 'model15905c093237_S2_stan_model' at line 60, column 16

```
58: vector<lower = 0>[M+1] rho;      // Mean fecundity
```

```
59: real<lower = 0> mu[M+1];
```

```
60: vector[M+1] mu;
```

I did not go into S3 in detail: it is long, seemingly repeats what is actually in the manuscript, and provides extra details for readers interested in the specifics of this study.

Providing the data did enable me to discover a major (critical) concern: the weather station data is dominated by missing values, and worse, there is a temporal pattern in the NAs (decreasing with time). The 'sum' rainfall is negatively correlated with the number of missing values. The current method thus invalidates the fledgling-rainfall analysis: in short, you cannot use the sum of rain. You will need to apply a randomisation procedure or something to correct for the missing value confounder, but I suspect some other rainfall measure e.g mean daily might be easier to use.

I ran this code in the Rain Prep section to uncover my concern (after the line reading the data and the code chunk creating the year variable, but before the final summary):

```
na_rain <- rain %>%
  group_by(year, id) %>%
  summarise(nans = sum(is.na(prcp))) %>%
  group_by(year) %>%
  summarise(nans = sum(nans, na.rm = T))
ggplot(na_rain, aes(year, nans))+geom_col()
```

There is a function conflict in the libraries provided which means that you need to use 'summarise' rather than 'summarize' if all libraries are run together as currently listed.

Minor Comments:

Abstract: "appropriate sitting". I guess that is placement of windfarm sites: I don't know how that transforms into a verb, but sitting just doesn't work.

A map of the sampling area would be useful area, especially to consider concerns regarding spatial autocorrelation.

Line 33: Are the italics required?

Methods:

Line 32: Clarify: The negative binomial assumption: is that yours or Royle and Nichols or the stats book reference? Maybe just start: The Royle-Nichols model further assumes....

Data selection: It is unclear what the spatial and temporal selection of pentads is. It reads like the entire range, but the species is migratory. Surely just the breeding range and temporal period is pertinent to this analysis (would also explicitly exclude the poorly covered Lesotho region).

I like the inclusion of the rainfall data.

Monte Carlo iterations: is 2000 enough? Maybe cite a precedent for this.

Mortality produced by windfarms. You may want to be explicit that you assume no behavioural adaptation or selection to the presence of turbines under this scenario.

Line 40: Mortality rates: just add some units here: are those 0-5 individuals per pentad per year? Or per range over some other timeframe? I found out later that is from the entire population.... So just clarify here to avoid confusion.

Results: This starts easily with the description of the data used, but then the second paragraph appears to be PVA results, leaving me wondering if I missed something regarding the occupancy modelling step, explained in great detail in the methods.

Discussion Pg 21, end of paragraph c Line 42: The oddity of 2014 was commented on before in results, so not sure if you need to repeat it so verbatim in the discussion. I wonder how much that had to do with the data management system, specifically the role of BirdLasser and the resulting repercussions in terms of reporting rate.

Review form: Reviewer 2 (Miguel Ferrer)

Is the manuscript scientifically sound in its present form?

No

Are the interpretations and conclusions justified by the results?

No

Is the language acceptable?

Yes

Do you have any ethical concerns with this paper?

No

Have you any concerns about statistical analyses in this paper?

Yes

Recommendation?

Reject

Comments to the Author(s)

Endangered raptor using detection/non-detection data reveals susceptibility to wind farm impacts

This is a draft trying to conduct an exercise of potential effect of additive mortality in a long-lived and low-fecundity bird of prey; the black harrier. Even if bird population has undergone a substantial decline in the last century; mainly due to anthropogenic activities like illegal shooting or accidents with man-made structures such as power lines, buildings or road network. In comparison to the effect of these structures, wind farms -induced disturbances on avian community are regularly considered negligible. Nevertheless, even small additional mortality can have a deleterious effect on a sensitive species. Using this well-known principle, authors conducted a simulation model to show that even a small theoretical increase in mortality would have strong consequences in a 100 years' period.

In my view, this approach is not a novel one, it is not providing any new insight in wind-farm and bird interactions or in harrier's population dynamics. Furthermore, this kind of approach have been demonstrated poorly accurate and even mostly wrong with the same topic and with raptor species as well. Scientists should be more careful when assessing risks from emerging threats to biodiversity on a large scale. Different examples showed how real populations' trajectories or impacts of human infrastructures on threatened species over the years might largely differ from published large-scale predictions. The Egyptian vulture in Spain, whose extinction in the Iberian Peninsula, due to mortality in wind farms, was predicted for nearly 2020, according published viability analyses (Carrete, M., Sánchez-Zapata, J. A., Benítez, J. R., Lobón, M., & Donázar, J. A. (2009). Large scale risk-assessment of wind-farms on population viability of a globally endangered long-lived raptor. *Biological Conservation*, 142(12), 2954-2961.), and yet, 20 years after this publication, not only it did not happen, but its national (and European) population remains stable and even slightly increasing (2.6%, Del Moral, J. C. y Molina, B. (Eds.) 2018. *El alimoche común en España, población reproductora en 2018 y método de censo*. SEO/BirdLife. Madrid). Nevertheless, this paper is cited by the authors a supporting their main conclusion. Taking a look to simulations output, specially to the huge confidence interval, it easy to understand that mean values should not be used as a good indicator of model predictions alone. This problem was the same in Carrete et al. paper.

Another concern is the long time used in simulations. 100 years is clearly too much for any accurate prediction, particularly when applied to a bird that can be very responsive to climate variations as the harriers. Others author had suggested the use of double the live-span of the species as better approach when simulate future scenarios (Morandini, V., Ferrer, M. 2017. How to plan reintroductions of long-lived birds? PLoS One 12:1-17).

Probably I misunderstood the explanation but it seems to me that you used a constant value of rain in every year of the simulation. Literally you said ". Since fecundity depends on rainfall, for the simulations we used the fecundity value that corresponds to the average rainfall observed in the studied period." This approach is limiting a lot the natural variability of the population's trajectories and I don't see why you have to used it.

Finally, I suggest the following papers closely related to the topic of this draft, but that seems to have been ignored. They included analyses of non-avoidance of harrier after installing wind farms, new approach trying to prevent and mitigate mortality of raptor at wind farms:

- Hernández-Pliego J, De Lucas M, Ferrer M, Muñoz A 2015. Effects of wind farms on Montagu's harrier (*Circus pygargus*) in southern Spain. *Biological Conservation* 191: 452-458.
- Ferrer, M., De Lucas, M., Janss, G.F.E., Casado, E., Muñoz, A.R., Bechard, M., Calabuig, C.P., 2012. Weak relationship between risk assessment studies and recorded mortality in wind farms. *Journal of Applied Ecology* 49: 38-46.
- De Lucas, M., Ferrer, M., Bechard, M., Muñoz, A.R. 2012. Griffon vulture mortality at wind farms in southern Spain: Distribution of fatalities and active mitigation measures. *Biological Conservation*, 147: 184-189.
- De Lucas, M., Ferrer, M., Janss, G.F.E. 2012. Using Wind Tunnels to Predict Bird Mortality in Wind Farms: The Case of Griffon Vultures. *Plos One* 7 (11): e48092. Q2

Decision letter (RSOS-210599.R0)

Dear Dr Cervantes Peralta

The Editors assigned to your paper RSOS-210599 "Population viability assessment of an Endangered raptor using detection/non-detection data reveals susceptibility to wind farm impacts" have made a decision based on their reading of the paper and any comments received from reviewers.

Regrettably, in view of the reports received, the manuscript has been rejected in its current form. However, a new manuscript may be submitted which takes into consideration these comments.

We invite you to respond to the comments supplied below and prepare a resubmission of your manuscript. Below the referees' and Editors' comments (where applicable) we provide additional requirements. We provide guidance below to help you prepare your revision.

Please note that resubmitting your manuscript does not guarantee eventual acceptance, and we do not generally allow multiple rounds of revision and resubmission, so we urge you to make every effort to fully address all of the comments at this stage. If deemed necessary by the Editors, your manuscript will be sent back to one or more of the original reviewers for assessment. If the original reviewers are not available, we may invite new reviewers.

Please resubmit your revised manuscript and required files (see below) no later than 02-Mar-2022. Note: the ScholarOne system will 'lock' if resubmission is attempted on or after this deadline. If you do not think you will be able to meet this deadline, please contact the editorial office immediately.

Please note article processing charges apply to papers accepted for publication in Royal Society Open Science (<https://royalsocietypublishing.org/rsos/charges>). Charges will also apply to papers transferred to the journal from other Royal Society Publishing journals, as well as papers submitted as part of our collaboration with the Royal Society of Chemistry (<https://royalsocietypublishing.org/rsos/chemistry>). Fee waivers are available but must be requested when you submit your manuscript (<https://royalsocietypublishing.org/rsos/waivers>).

Thank you for submitting your manuscript to Royal Society Open Science and we look forward to receiving your resubmission. If you have any questions at all, please do not hesitate to get in touch.

on behalf of Professor Leslie Brown (Associate Editor) and Pete Smith (Subject Editor)
openscience@royalsociety.org

Associate Editor Comments to Author (Professor Leslie Brown):

Associate Editor: 1

Comments to the Author:

Dear authors, thank you for submitting your manuscript to our journal. Based on the reviewers comments there are serious flaws in the manuscript especially with respect to the data analyses and raw data. The huge confidence interval in the data analyses is also an issue in that it indicates that mean values should not be used as a good indicator of model predictions alone. In addition the weather station data has many missing values while the one reviewer could not run the same analyses/simulations as you using your raw dataset. It also seems as though you only take the wind farms into consideration as the only anthropogenic influences that causes a decline of the species, while there are many other factors contributing too. I have therefore recommended that the manuscript in its current format be rejected and advise that you revise it addressing the different concerns and resubmit it for evaluation.

Associate Editor: 2

Comments to the Author:

Dear authors, thank you for submitting your manuscript. I have requested that it be sent for reviewing.

Reviewer comments to Author:

Reviewer: 1

Comments to the Author(s)

Black Harriers are Endangered raptors that are endemic to southern Africa. This paper examines the population viability given inevitable wind infrastructure developments across the range. Use is made of the SABAP2 citizen science project. The use of this data to conduct a PVA is especially novel. The paper is topical, current, well written, and I hope it has the intended impact of encouraging conservation of this beautiful bird of prey. While framed in the light of potential wind energy impacts, realistically one could replace wind energy with any negative impacts.

I base my review on pages 14 – 25 of the pdf provided, since this seems to include in text references etc, the previous pages might be an old version, and the purpose of it being rendered in the pdf was not clear to me. I also do not like numeric citation style for the purposes of review (completely unreasonable to follow each link), and so I make no comment on the pertinence of all selected in-situ references, which I could have done with Author Date style: that is critique is on the journal. The few references I checked seemed fine.

The extent of the supplementary information is extraordinary and exemplary in terms of tidiness and succinctness: it is beautiful code. However, it fails reproducibility. I managed to reproduce code given the data provided until the stan model, when things failed and I was unable to proceed despite much time spent at attempting debugging on my side. This was the persistent error message:

SYNTAX ERROR, MESSAGE(S) FROM PARSER:

Duplicate declaration of variable, name=mu; attempt to redeclare as vector in transformed parameter; previously declared as real[] in transformed parameter

error in 'model15905c093237_S2_stan_model' at line 60, column 16

```
58: vector[M+1] rho;      // Mean fecundity
59: real mu[M+1];
60: vector[M+1] mu;
```

I did not go into S3 in detail: it is long, seemingly repeats what is actually in the manuscript, and provides extra details for readers interested in the specifics of this study.

Providing the data did enable me to discover a major (critical) concern: the weather station data is dominated by missing values, and worse, there is a temporal pattern in the NAs (decreasing with time). The 'sum' rainfall is negatively correlated with the number of missing values. The current method thus invalidates the fledgling-rainfall analysis: in short, you cannot use the sum of rain. You will need to apply a randomisation procedure or something to correct for the missing value confounder, but I suspect some other rainfall measure e.g mean daily might be easier to use.

I ran this code in the Rain Prep section to uncover my concern (after the line reading the data and the code chunk creating the year variable, but before the final summary):

```
na_rain <- rain %>%
  group_by(year, id) %>%
  summarise(nans = sum(is.na(prcp))) %>%
  group_by(year) %>%
  summarise(nans = sum(nans, na.rm = T))
ggplot(na_rain, aes(year, nans))+geom_col()
```

There is a function conflict in in the libraries provided which means that you need to use 'summarise' rather than 'summarize' if all libraries are run together as currently listed.

Minor Comments:

Abstract: "appropriate sitting". I guess that is placement of windfarm sites: I don't know how that transforms into a verb, but sitting just doesn't work.

A map of the sampling area would be useful area, especially to consider concerns regarding spatial autocorrelation.

Line 33: Are the italics required?

Methods:

Line 32: Clarify: The negative binomial assumption: is that yours or Royle and Nichols or the stats book reference? Maybe just start: The Royle-Nichols model further assumes....

Data selection: It is unclear what the spatial and temporal selection of pentads is. It reads like the entire range, but the species is migratory. Surely just the breeding range and temporal period is pertinent to this analysis (would also explicitly exclude the poorly covered Lesotho region).

I like the inclusion of the rainfall data.

Monte Carlo iterations: is 2000 enough? Maybe cite a precedent for this.

Mortality produced by windfarms. You may want to be explicit that you assume no behavioural adaptation or selection to the presence of turbines under this scenario.

Line 40: Mortality rates: just add some units here: are those 0-5 individuals per pentad per year? Or per range over some other timeframe? I found out later that is from the entire population.... So just clarify here to avoid confusion.

Results: This starts easily with the description of the data used, but then the second paragraph appears to be PVA results, leaving me wondering if I missed something regarding the occupancy modelling step, explained in great detail in the methods.

Discussion Pg 21, end of paragraph c Line 42: The oddity of 2014 was commented on before in results, so not sure if you need to repeat it so verbatim in the discussion. I wonder how much that had to do with the data management system, specifically the role of BirdLasser and the resulting repercussions in terms of reporting rate.

Reviewer: 2

Comments to the Author(s)

Endangered raptor using detection/non-detection data reveals susceptibility to wind farm impacts

This is a draft trying to conduct an exercise of potential effect of additive mortality in a long-lived and low-fecundity bird of prey; the black harrier. Even if bird population has undergone a substantial decline in the last century; mainly due to anthropogenic activities like illegal shooting or accidents with man-made structures such as power lines, buildings or road network. In

comparison to the effect of these structures, wind farms –induced disturbances on avian community are regularly considered negligible. Nevertheless, even small additional mortality can have a deleterious effect on a sensitive species. Using this well-known principle, authors conducted a simulation model to show that even a small theoretical increase in mortality would have strong consequences in a 100 years' period.

In my view, this approach is not a novel one, it is not providing any new insight in wind-farm and bird interactions or in harrier's population dynamics. Furthermore, this kind of approach have been demonstrated poorly accurate and even mostly wrong with the same topic and with raptor species as well. Scientists should be more careful when assessing risks from emerging threats to biodiversity on a large scale. Different examples showed how real populations' trajectories or impacts of human infrastructures on threatened species over the years might largely differ from published large-scale predictions. The Egyptian vulture in Spain, whose extinction in the Iberian Peninsula, due to mortality in wind farms, was predicted for nearly 2020, according published viability analyses (Carrete, M., Sánchez-Zapata, J. A., Benítez, J. R., Lobón, M., & Donázar, J. A. (2009). Large scale risk-assessment of wind-farms on population viability of a globally endangered long-lived raptor. *Biological Conservation*, 142(12), 2954-2961.), and yet, 20 years after this publication, not only it did not happen, but its national (and European) population remains stable and even slightly increasing (2.6%, Del Moral, J. C. y Molina, B. (Eds.) 2018. *El alimoche común en España, población reproductora en 2018 y método de censo*. SEO/BirdLife. Madrid). Nevertheless, this paper is cited by the authors a supporting their main conclusion. Taking a look to simulations output, specially to the huge confidence interval, it easy to understand that mean values should not be used as a good indicator of model predictions alone. This problem was the same in Carrete et al. paper.

Another concern is the long time used in simulations. 100 years is clearly too much for any accurate prediction, particularly when applied to a bird that can be very responsive to climate variations as the harriers. Others author had suggested the use of double the live-span of the species as better approach when simulate future scenarios (Morandini, V., Ferrer, M. 2017. How to plan reintroductions of long-lived birds? *PLoS One* 12:1-17).

Probably I misunderstood the explanation but it seems to me that you used a constant value of rain in every year of the simulation. Literally you said ". Since fecundity depends on rainfall, for the simulations we used the fecundity value that corresponds to the average rainfall observed in the studied period." This approach is limiting a lot the natural variability of the population's trajectories and I don't see why you have to used it.

Finally, I suggest the following papers closely related to the topic of this draft, but that seems to have been ignored. They included analyses of non-avoidance of harrier after installing wind farms, new approach trying to prevent and mitigate mortality of raptor at wind farms:

- Hernández-Pliego J, De Lucas M, Ferrer M, Muñoz A 2015. Effects of wind farms on Montagu's harrier (*Circus pygargus*) in southern Spain. *Biological Conservation* 191: 452-458.
- Ferrer, M., De Lucas, M., Janss, G.F.E., Casado, E., Muñoz, A.R., Bechard, M., Calabuig, C.P., 2012. Weak relationship between risk assessment studies and recorded mortality in wind farms. *Journal of Applied Ecology* 49: 38-46.
- De Lucas, M., Ferrer, M., Bechard, M., Muñoz, A.R. 2012. Griffon vulture mortality at wind farms in southern Spain: Distribution of fatalities and active mitigation measures. *Biological Conservation*, 147: 184-189.
- De Lucas, M., Ferrer, M., Janss, G.F.E. 2012. Using Wind Tunnels to Predict Bird Mortality in Wind Farms: The Case of Griffon Vultures. *Plos One* 7 (11): e48092. Q2

===PREPARING YOUR MANUSCRIPT===

one version identifying all the changes that have been made (for instance, in coloured highlight, in bold text, or tracked changes);
 a 'clean' version of the new manuscript that incorporates the changes made, but does not highlight them. This version will be used for typesetting if your manuscript is accepted.

===PREPARING YOUR REVISION IN SCHOLARONE===

- Any electronic supplementary material (ESM).
- If you are requesting a discretionary waiver for the article processing charge, the waiver form must be included at this step.
- If you are providing image files for potential cover images, please upload these at this step, and inform the editorial office you have done so. You must hold the copyright to any image provided.
- A copy of your point-by-point response to referees and Editors. This will expedite the preparation of your proof.

- Ensure that your data access statement meets the requirements at <https://royalsociety.org/journals/authors/author-guidelines/#data>. You should ensure that you cite the dataset in your reference list. If you have deposited data etc in the Dryad repository, please include both the 'For publication' link and 'For review' link at this stage.
- If you are requesting an article processing charge waiver, you must select the relevant waiver option (if requesting a discretionary waiver, the form should have been uploaded at Step 3 'File upload' above).
- If you have uploaded ESM files, please ensure you follow the guidance at <https://royalsociety.org/journals/authors/author-guidelines/#supplementary-material> to include a suitable title and informative caption. An example of appropriate titling and captioning may be found at https://figshare.com/articles/Table_S2_from_Is_there_a_trade-off_between_peak_performance_and_performance_breadth_across_temperatures_for_aerobic_scope_in_teleost_fishes_/3843624.

Author's Response to Decision Letter for (RSOS-210599.R0)

See Appendix A.

Decision letter (RSOS-220043.R0)

Dear Dr Cervantes Peralta

On behalf of the Editors, we are pleased to inform you that your Manuscript RSOS-220043 "Population viability assessment of an Endangered raptor using detection/non-detection data reveals susceptibility to anthropogenic impacts" has been accepted for publication in Royal Society Open Science subject to minor revision in accordance with the referees' reports. Please find the referees' comments along with any feedback from the Editors below my signature.

Please submit your revised manuscript and required files (see below) no later than 7 days from today's (ie 24-Jan-2022) date. Note: the ScholarOne system will 'lock' if submission of the revision is attempted 7 or more days after the deadline. If you do not think you will be able to meet this deadline please contact the editorial office immediately.

on behalf of Professor Leslie Brown (Associate Editor) and Pete Smith (Subject Editor)
openscience@royalsociety.org

Associate Editor Comments to Author (Professor Leslie Brown):
Comments to the Author:

Thank you for your resubmission and detailed responses to the reviewers' comments. I am satisfied that you have sufficiently addressed the various concerns raised by the reviewers and that you have clearly stated the constraints of your results and data. I have read through some sections and noted only a few typo errors e.g. line 264 "harriers" should be "harrier" and 296 "Black Harrier" should be "Black Harriers". Please go through the document and ensure that all errors have been corrected. Also consider changing "here" in line 302 to "in this study".

===PREPARING YOUR MANUSCRIPT===

one version should clearly identify all the changes that have been made (for instance, in coloured highlight, in bold text, or tracked changes);
a 'clean' version of the new manuscript that incorporates the changes made, but does not highlight them. This version will be used for typesetting.

===PREPARING YOUR REVISION IN SCHOLARONE===

-- If you are requesting an article processing charge waiver, you must select the relevant waiver option (if requesting a discretionary waiver, the form should have been uploaded, see 'File upload' above).

-- If you have uploaded any electronic supplementary (ESM) files, please ensure you follow the guidance at <https://royalsociety.org/journals/authors/author-guidelines/#supplementary-material> to include a suitable title and informative caption. An example of appropriate titling and captioning may be found at https://figshare.com/articles/Table_S2_from_Is_there_a_trade-off_between_peak_performance_and_performance_breadth_across_temperatures_for_aerobic_scope_in_teleost_fishes_/3843624.

Author's Response to Decision Letter for (RSOS-220043.R0)

See Appendix B.

Decision letter (RSOS-220043.R1)

Dear Dr Cervantes Peralta,

I am pleased to inform you that your manuscript entitled "Population viability assessment of an Endangered raptor using detection/non-detection data reveals susceptibility to anthropogenic impacts" is now accepted for publication in Royal Society Open Science.

Please remember to make any data sets or code libraries 'live' prior to publication, and update any links as needed when you receive a proof to check - for instance, from a private 'for review'

URL to a publicly accessible 'for publication' URL. It is good practice to also add data sets, code and other digital materials to your reference list.

on behalf of Professor Leslie Brown (Associate Editor) and Pete Smith (Subject Editor)
openscience@royalsociety.org

Appendix A

Dear editors,

We kindly thank you and reviewers for the comments on the manuscript, which proved useful in improving our analysis, as well as tightening our wording and arguments.

We have tried to address all the concerns to the best of our abilities, and we hope you agree with us that the manuscript is now more accurate, balanced and clear.

Please, see our inline responses to the reviewers below, in italics.

Sincerely,

Francisco Cervantes, Marlei Martins and Rob Simmons

Associate Editor Comments to Author (Professor Leslie Brown):

Associate Editor: 1 Comments to the Author:

Dear authors, thank you for submitting your manuscript to our journal. Based on the reviewers comments there are serious flaws in the manuscript especially with respect to the data analyses and raw data. The huge confidence interval in the data analyses is also an issue in that it indicates that mean values should not be used as a good indicator of model predictions alone. In addition the weather station data has many missing values while the one reviewer could not run the same analyses/simulations as you using your raw dataset. It also seems as though you only take the wind farms into consideration as the only anthropogenic influences that causes a decline of the species, while there are many other factors contributing too. I have therefore recommended that the manuscript in its current format be rejected and advise that you revise it addressing the different concerns and resubmit it for evaluation.

Associate Editor: 2 Comments to the Author:

Dear authors, thank you for submitting your manuscript. I have requested that it be sent for reviewing.

Reviewer comments to Author:

Reviewer: 1 Comments to the Author(s)

Black Harriers are Endangered raptors that are endemic to southern Africa. This paper examines the population viability given inevitable wind infrastructure developments across the range. Use is made of the SABAP2 citizen science project. The use of this data to conduct a PVA is especially novel. The paper is topical, current, well written, and I hope it has the intended impact of encouraging conservation of this beautiful bird of prey. While framed in the light of potential wind energy impacts, realistically one could replace wind energy with any negative impacts.

- *Thank you, we are happy to see that the topic is considered relevant. We totally agree with the view that our analysis could be applied to any kind of additional mortality, not only wind farms. We do mention this in the discussion. However, we also feel that at the moment wind energy is a new threat to a species that traditionally has not been particularly affected by human-associated mortality. Habitat loss, pesticides and power lines are all factors that combine to reduce or compromise the harrier population. Ways to reverse these have been treated in Curtis et al. 2004 and García-Heras et al. 2018, and that is why we focus on the new and contemporary threat of wind farms over and above the (known) threats. As such it is a novel threat and requires a novel analysis. We do not discount the severity of the habitat loss, so the review is correct to raise this. We have added two lines in the introduction to make this clear (see lines 35-37). However, we have also changed the narrative at several points in the manuscript and the title to reflect the fact that, indeed, additional mortality from any source could have the same effects.*

I base my review on pages 14 – 25 of the pdf provided, since this seems to include in text references etc, the previous pages might be an old version, and the purpose of it being rendered in the pdf was not clear to me. I also do not like numeric citation style for the purposes of review (completely unreasonable to follow each link), and so I make no comment on the pertinence of all selected in-situ references, which I could have done with Author Date style: that is critique is on the journal. The few references I checked seemed fine.

The extent of the supplementary information is extraordinary and exemplary in terms of tidiness and succinctness: it is beautiful code. However, it fails reproducibility. I managed to reproduce code given the data provided until the stan model, when things failed and I was unable to proceed despite much time spent at attempting debugging on my side. This was the persistent error message:

SYNTAX ERROR, MESSAGE(S) FROM PARSER:

Duplicate declaration of variable, name= μ ; attempt to redeclare as vector in transformed parameter; previously declared as real[] in transformed parameter

error in 'model15905c093237_S2_stan_model' at line 60, column 16

58: vector[M+1] rho; // Mean fecundity

59: real mu[M+1];

60: vector[M+1] mu;

- *Apologies for the double declaration of variables. The Stan code has been fixed.*

I did not go into S3 in detail: it is long, seemingly repeats what is actually in the manuscript, and provides extra details for readers interested in the specifics of this study.

Providing the data did enable me to discover a major (critical) concern: the weather station data is dominated by missing values, and worse, there is a temporal pattern in the NAs (decreasing with

time). The ‘sum’ rainfall is negatively correlated with the number of missing values. The current method thus invalidates the fledgling-rainfall analysis: in short, you cannot use the sum of rain. You will need to apply a randomisation procedure or something to correct for the missing value confounder, but I suspect some other rainfall measure e.g mean daily might be easier to use.

I ran this code in the Rain Prep section to uncover my concern (after the line reading the data and the code chunk creating the year variable, but before the final summary):

```
na_rain <- rain %>%  
  group_by(year, id) %>%  
  summarise(nans = sum(is.na(prcp))) %>%  
  group_by(year) %>%  
  summarise(nans = sum(nans, na.rm = T))  
ggplot(na_rain, aes(year, nans))+geom_col()
```

- *Thanks very much for picking this up. We have assessed the number of missing values per year and it does indeed grow over the years, from ca 40% to ca 50%. We have addressed this issue by removing the missing values and taking the average rainfall instead of the accumulated rainfall. This should take care of the different number of values in different years. The results of the analysis have not changed substantially. In fact, Black Harrier reporting rates correlated even better with the average rainfall than they did with the accumulated rainfall. Thus, further supporting our previous findings.*

There is a function conflict in in the libraries provided which means that you need to use ‘summarise’ rather than ‘summarize’ if all libraries are run together as currently listed.

- *We have explicitly declared from what package functions come from, using the construct `package::function`.*

Minor Comments:

Abstract: “appropriate sitting”. I guess that is placement of windfarm sites: I don’t know how that transforms into a verb, but sitting just doesn’t work.

- *Apologies, that should have read “siting” We will use “placement” instead*

A map of the sampling area would be useful area, especially to consider concerns regarding spatial autocorrelation.

- *We have added a figure with the pentads that were analysed.*

Line 33: Are the italics required?

- *This is a literal quote, but we have removed the italics because it is only a couple of words*

Methods:

Line 32: Clarify: The negative binomial assumption: is that yours or Royle and Nichols or the stats book reference? Maybe just start: The Royle-Nichols model further assumes....

- *There is no prescribed distribution for this model, so we used negative binomial to allow extra flexibility in mean-variance relationship. We have rephrased this section and it now makes it clear that this was our choice (lines 75 to 77).*

Data selection: It is unclear what the spatial and temporal selection of pentads is. It reads like the entire range, but the species is migratory. Surely just the breeding range and temporal period is pertinent to this analysis (would also explicitly exclude the poorly covered Lesotho region).

- *It is the entire range between 2008 and 2019. We have added a sentence to make this clear. Not all Black Harriers migrate, and notably not young birds, so excluding the non-breeding period could potentially exclude an important part of the population and could bias survival probabilities towards adult values. On the other hand, we admit that we might be overestimating the population of Black Harriers because the same harrier could be detected twice: once during summer and once during winter. However, we believe that the estimate of the rate at which population changes should not be affected by this fact as long as the selection of pentads remains constant over time. For our purposes, the rate of change is more important than a precise estimate of the population. We clarify this in the discussion.*

I like the inclusion of the rainfall data.

Monte Carlo iterations: is 2000 enough? Maybe cite a precedent for this.

- *We understand this comment refers to the Hamiltonian Monte Carlo samples. The important thing to look at are the effective number of samples, provided in table 1. There is no prescribed number of samples considered enough, but one can think of effective samples being independent and think about whether such a number could be considered enough to capture certain characteristics of the distribution. Convergence is also important, and we provide the Rhat statistic which should be close to 1. Gelman et al (2014) give some indication of the use of both effective sample size and Rhat, and we have added that reference.*

Mortality produced by windfarms. You may want to be explicit that you assume no behavioural adaptation or selection to the presence of turbines under this scenario.

- *We have added a sentence to the discussion to make this clear. Indeed, unpublished GPS tracking data shows no avoidance of wind farms by harriers.*

Line 40: Mortality rates: just add some units here: are those 0-5 individuals per pentad per year? Or per range over some other timeframe? I found out later that is from the entire population.... So just clarify here to avoid confusion.

- *Clarified, thank you!*

Results: This starts easily with the description of the data used, but then the second paragraph appears to be PVA results, leaving me wondering if I missed something regarding the occupancy modelling step, explained in great detail in the methods.

- *We don't really model occupancy. We use detection/non-detection data to estimate the rate of change and life-history parameters of the population. The only thing that we estimate related to occupancy is the detection probability.*

Discussion Pg 21, end of paragraph c Line 42: The oddity of 2014 was commented on before in results, so not sure if you need to repeat it so verbatim in the discussion. I wonder how much that had to do with the data management system, specifically the role of BirdLasser and the resulting repercussions in terms of reporting rate.

- *Good point, not sure about that, but it seems clear that it is an outlier difficult to explain. If it were BirdLasser, then the bias should be consistent after 2014 to 2018. But it was only 2014, so it must be some other variable. We just wanted to be clear about potential limitations of our model. With the new approach of using average precipitation values this data point is better explained by the model, but still out of the 95% credible region.*

Reviewer: 2

Comments to the Author(s)

Endangered raptor using detection/non-detection data reveals susceptibility to wind farm impacts

This is a draft trying to conduct an exercise of potential effect of additive mortality in a long-lived and low-fecundity bird of prey; the black harrier. Even if bird population has undergone a substantial decline in the last century; mainly due to anthropogenic activities like illegal shooting or accidents with man-made structures such as power lines, buildings or road network.

In comparison to the effect of these structures, wind farms –induced disturbances on avian community are regularly considered negligible. Nevertheless, even small additional mortality can have a deleterious effect on a sensitive species. Using this well-known principle, authors conducted a simulation model to show that even a small theoretical increase in mortality would have strong consequences in a 100 years' period.

- *The main driver for population decline of the Black Harrier has been habitat loss, produced largely by the transformation of indigenous Fynbos and Renosterveld habitat into agricultural land. Wind energy is a new, contemporary threat to a species that traditionally has not been particularly affected by (direct) human-associated mortality. There is no evidence that illegal shooting or accidents with man-made structures such as buildings has played any role. We would only agree that perhaps power lines are a concern in this respect, and new energy infrastructure comes associated with the installation of new transmission lines. Therefore, this new form of mortality could add a new dimension to the management of the species. We have added two lines in the introduction to make this clear (see lines 35-37).*

In my view, this approach is not a novel one, it is not providing any new insight in wind-farm and bird interactions or in harrier's population dynamics.

- *We agree that the fact that long-lived species with slow life-histories are sensitive to additional mortality is not new. Our aim in this manuscript was not to prove this point, but rather to i) use detection/non-detection data to estimate life-history parameters and population's rate of change for a species – this methodology is novel, as far as we are*

aware, at least in the context of wind energy planning -, and ii) provide a quantitative framework for the viability of the Black Harrier (how many fatalities are too many). The Black Harrier as an Endangered species has also never been subject to Population Viability Assessment, so this kind of quantitative assessment is also novel. The fact that a species is “sensitive” is quite vague, and we are often faced with the question of how many individuals can a species lose before its population is severely impacted (e.g., for Environmental Impact Assessments and management plans). We have modified lines 48-51 in the introduction to make this clearer.

Furthermore, this kind of approach have been demonstrated poorly accurate and even mostly wrong with the same topic and with raptor species as well. Scientists should be more careful when assessing risks from emerging threats to biodiversity on a large scale. Different examples showed how real populations’ trajectories or impacts of human infrastructures on threatened species over the years might largely differ from published large-scale predictions. The Egyptian vulture in Spain, whose extinction in the Iberian Peninsula, due to mortality in wind farms, was predicted for nearly 2020, according published viability analyses (Carrete, M., Sánchez-Zapata, J. A., Benítez, J. R., Lobón, M., & Donázar, J. A. (2009). Large scale risk-assessment of wind-farms on population viability of a globally endangered long-lived raptor. *Biological Conservation*, 142(12), 2954-2961.), and yet, 20 years after this publication, not only it did not happen, but its national (and European) population remains stable and even slightly increasing (2.6%, Del Moral, J. C. y Molina, B. (Eds.) 2018. *El alimoche común en España, población reproductora en 2018 y método de censo*. SEO/BirdLife. Madrid). Nevertheless, this paper is cited by the authors a supporting their main conclusion.

- *We understand that “this kind of approach” refers to “population viability assessments”, and we argue that there are different ways of approaching these problems. Also, we are not particularly versed on the conservation situation of the Egyptian Vultures, a species that has, most likely, already gone extinct in southern Africa. Therefore, we feel we should not embark on the discussion of someone else’s work. We would also not want to compare a scavenging species with an active predator like the Black Harrier, because conservation problems of these might be very different. For example, we could argue that vultures, as scavengers, are highly susceptible to poison, because this is the main reason for their demise in southern Africa, and possibly in Spain too. Thus, if the main threat of poisons has been treated in Spain, this could explain the observed population rebound, even in presence of wind farm impacts. But as a recent and fast-developing industry in South Africa, already known to kill Black Harriers at a relatively high rate (and more so than in Europe) this is likely to be a significant threat to a species numbering a mere 1300 individuals (two magnitudes lower than any harrier species in Europe!).*
- *However, we do understand that prediction of population trajectories is difficult, and models are subject to assumptions and limitations (this is also acknowledged by Carrete et al.) that must be clearly stated. That is one reason why we do not advocate for the use of out-of-the-box solutions such as Vortex (used by Carrete et al., and many others). Instead, we prefer specifying bespoke models for each situation. We have tried our best to follow both principles and specify a bespoke and transparent model. We are keen to receive feedback on any specific concerns by the reviewers, so that we can improve the model, if necessary.*

Taking a look to simulations output, specially to the huge confidence interval, it easy to understand that mean values should not be used as a good indicator of model predictions alone. This problem was the same in Carrete et al. Paper.

- *These are not confidence intervals, but Bayesian credible intervals. More precisely, the different colours depict credible regions that represent the probability of the “true” parameter being in those intervals. These can therefore be interpreted as the probability distribution of the predictions, as opposed to classic confidence intervals, allowing us to better inform uncertainty in our predictions. This is implied in the fact that simulations are based on samples from the model posterior distribution (this is explained in lines 160-161).*
- *By mean population we meant population trajectories predicted using the mean rainfall values observed for the period. We should have been more clear, but now we have changed our approach and we are not using mean rainfall values (see below), so we are no longer using this terminology. Similarly, to estimate the probability of extinction, we don't use the mean of the predictions, instead we take the proportion of posterior predictions that are zero or smaller. We have added a sentence in line 176 to make this clear.*
- *It seems to us that Carrete et al., did not provide uncertainty estimates for their predicted population size. We wanted to be transparent in this sense, and therefore provided credible regions. We further state in our discussion (lines 230-232) that the wide range of the credible regions reflect the difficulty of predicting population trajectories, particularly from detection/non-detection data.*

Another concern is the long time used in simulations. 100 years is clearly too much for any accurate prediction, particularly when applied to a bird that can be very responsive to climate variations as the harriers. Others author had suggested the use of double the liv[f]e-span of the species as better approach when simulate future scenarios (Morandini, V., Ferrer, M. 2017. How to plan reintroductions of long-lived birds? PLoS One 12:1-17).

- *We agree that predictions 100 years ahead may not be accurate, and we have tried to be transparent with our uncertainty estimates. The most important estimate is the rate of change of the population. This is not affected by the horizon of the predictions but rather by the estimated life-history parameters and additional mortality. We selected this time horizon just to be able to estimate extinction probabilities to compare the different cases. We have tried to clarify this further in the discussion (lines 232 to 234 and 249-252). Note that the International Union for the Conservation of Nature (IUCN) uses 100-year threshold in their viability assessments for species red listing, so our assessment is in line with international standards.*
- *By no means do we think that environmental conditions will not change in the future, but there is a need to develop management strategies now, so we have taken projections based on the current situation. We have tried to make this clear in the introduction (lines 49-51). Furthermore, the Inter-governmental Panel on Climate Change can only loosely predict how climate is going to change in the next 100 years. However, their forecast of a reduction in rainfall for southern Africa is likely to affect the Black Harrier negatively, if the relationship between its fecundity and rainfall is causative (this is discussed in the manuscript, lines 254-260).*

Probably I misunderstood the explanation but it seems to me that you used a constant value of rain in every year of the simulation. Literally you said “ . Since fecundity depends on rainfall, for the simulations we used the fecundity value that corresponds to the average rainfall observed in the studied period.” This approach is limiting a lot the natural variability of the population’s trajectories and I don’t see why you have to use it.

- *We used the mean value of rainfall for simplicity, because we can’t predict what rainfall will be in the future. However, we recognize that this is a fair point that might be raised by other readers. Therefore, we have sampled rainfall values from the data at each step of the simulations instead of using a fixed average. Thus, we demonstrate how, although more rugged, the results are similar and the predictive range narrower. This is possibly because the mean value is sensitive to outliers and particularly wet years could be driving the mean upwards, while the general pattern of rainfall is downwards.*

Finally, I suggest the following papers closely related to the topic of this draft, but that seems to have been ignored. They included analyses of non-avoidance of harrier after installing wind farms, new approach trying to prevent and mitigate mortality of raptor at wind farms:

- *Thanks for providing these references. Although are all great bird and wind energy references, due to limited space, we need to be selective with what references we include. We comment below.*
- Hernández-Pliego J, De Lucas M, Ferrer M, Muñoz A 2015. Effects of wind farms on Montagu's harrier (*Circus pygargus*) in southern Spain. *Biological Conservation* 191: 452-458.
 - *This is a very relevant reference indeed, and we have added it (line 27).*
- Ferrer, M., De Lucas, M., Janss, G.F.E., Casado, E., Muñoz, A.R., Bechard, M., Calabuig, C.P., 2012. Weak relationship between risk assessment studies and recorded mortality in wind farms. *Journal of Applied Ecology* 49: 38-46.
 - *This reference deals with pre-construction monitoring and environmental impact assessment, rather than with strategic planning, or population-level effects, so we have not added it.*
- De Lucas, M., Ferrer, M., Bechard, M., Muñoz, A.R. 2012. Griffon vulture mortality at wind farms in southern Spain: Distribution of fatalities and active mitigation measures. *Biological Conservation*, 147: 184-189.
 - *This reference talks about mitigation measures for vultures, which could not apply to harriers. We do not argue that fatalities can be prevented, we discuss what could happen if they are not. Therefore, we have not included this reference.*
- De Lucas, M., Ferrer, M., Janss, G.F.E. 2012. Using Wind Tunnels to Predict Bird Mortality in Wind Farms: The Case of Griffon Vultures. *Plos One* 7 (11): e48092. Q2
 - *This paper deals with ways of predicting where vulture flights might occur. Again, we do not argue that wind farm impacts can be mitigated, we provide the means to estimate what can happen if mortality does happen and provide guidance on an important species, the Black Harrier. Also, harriers are very different from the social and scavenging vultures, and*

measures design for the latter, may not be useful for solitary and predatory harriers. We have not included this reference.

Appendix B

Dear editors,

We are very pleased and thankful for the decision to accept the manuscript.

We have revised the text and made some minor changes with the objective of improving the grammar and readability of the paper. We have also picked up a mistake in the results section, in which the extinction probabilities were not updated from the first version. All tables, figures, and discussion reflected and took into consideration the correct values, but the text in the results section was not updated, for some reason. The text has now been fixed, and everything is coherent.

We hope the paper is now up to standard. We appreciate the time and effort put into reviewing this paper, which has benefited its overall quality.

Sincerely,

Francisco Cervantes, Marlei Martins and Rob Simmons